# Conditionally Adaptive Multi-Task Learning: Improving Transfer Learning in NLP Using Fewer Parameters & Less Data

**Jonathan Pilault**[1]*, **Amine El hattami**[1]*, **Christopher Pal**[1,2,3]
[1]Polytechnique Montreal & Mila, [2]Element AI, [3]Canada CIFAR AI Chair
`{jonathan.pilault,amine.elhattami,christopher.pal}@polymtl.ca`

## Abstract

Multi-Task Learning (MTL) networks have emerged as a promising method for transferring learned knowledge across different tasks. However, MTL must deal with challenges such as: overfitting to low resource tasks, catastrophic forgetting, and negative task transfer, or learning interference. Often, in Natural Language Processing (NLP), a separate model per task is needed to obtain the best performance. However, many fine-tuning approaches are both parameter inefficient, i.e., potentially involving one new model per task, and highly susceptible to losing knowledge acquired during pretraining. We propose a novel Transformer based Adapter consisting of a new conditional attention mechanism as well as a set of task-conditioned modules that facilitate weight sharing. Through this construction, we achieve more efficient parameter sharing and mitigate forgetting by keeping half of the weights of a pretrained model fixed. We also use a new multi-task data sampling strategy to mitigate the negative effects of data imbalance across tasks. Using this approach, we are able to surpass single task fine-tuning methods while being parameter and data efficient (using around 66% of the data for weight updates). Compared to other BERT Large methods on GLUE, our 8-task model surpasses other Adapter methods by 2.8% and our 24-task model outperforms by 0.7-1.0% models that use MTL and single task fine-tuning. We show that a larger variant of our single multi-task model approach performs competitively across 26 NLP tasks and yields state-of-the-art results on a number of test and development sets. Our code is publicly available at `https://github.com/CAMTL/CA-MTL`.

## 1 Introduction

The introduction of deep, contextualized Masked Language Models (MLM)[1] trained on massive amounts of unlabeled data has led to significant advances across many different Natural Language Processing (NLP) tasks (Peters et al., 2018; Liu et al., 2019a). Much of these recent advances can be attributed to the now well-known BERT approach (Devlin et al., 2018). Substantial improvements over previous state-of-the-art results on the GLUE benchmark (Wang et al., 2018) have been obtained by multiple groups using BERT models with task specific fine-tuning. The "BERT-variant + fine-tuning" formula has continued to improve over time with newer work constantly pushing the state-of-the-art forward on the GLUE benchmark. The use of a single neural architecture for multiple NLP tasks has shown promise long before the current wave of BERT inspired methods (Collobert & Weston, 2008) and recent work has argued that autoregressive language models (ARLMs) trained on large-scale datasets – such as the GPT family of models (Radford et al., 2018), are in practice multi-task learners (Brown et al., 2020). However, even with MLMs and ARLMs trained for multi-tasking, single task fine-tuning is usually also employed to achieve state-of-the-art performance on specific tasks of interest. Typically this fine-tuning process may entail: creating a task-specific fine-tuned model (Devlin et al., 2018), training specialized model components for task-specific predictions (Houlsby et al., 2019) or fine-tuning a single multi-task architecture (Liu et al., 2019b).

---

*Joint first-authors

[1]For reader convenience, all acronyms in this paper are summarized in section A.1 of the Appendix.

*Single-task* fine-tuning overall pretrained model parameters may have other issues. Recent analyses of such MLM have shed light on the linguistic knowledge that is captured in the hidden states and attention maps (Clark et al., 2019b; Tenney et al., 2019a; Merchant et al., 2020). Particularly, BERT has middle Transformer (Vaswani et al., 2017) layers that are typically the most transferable to a downstream task (Liu et al., 2019a). The model proxies the steps of the traditional NLP pipeline in a localizable way (Tenney et al., 2019a) — with basic syntactic information appearing earlier in the network, while high-level semantic information appearing in higher-level layers. Since pretraining is usually done on large-scale datasets, it may be useful, for a variety of downstream tasks, to conserve that knowledge. However, single task fine-tuning causes catastrophic forgetting of the knowledge learned during MLM (Howard & Ruder, 2018). To preserve knowledge, freezing part of a pretrained network and using *Adapters* for new tasks have shown promising results (Houlsby et al., 2019).

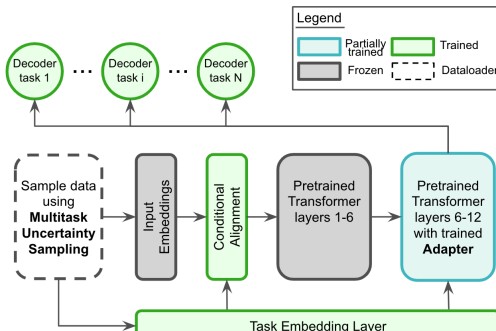

Figure 1: CA-MTL base architecture with our uncertainty-based sampling algorithm. Each task has its own decoder. The input embedding layer and the lower Transformer layers are frozen. The upper Transformer layer and Conditional Alignment module are modulated with the task embedding.

Inspired by the human ability to transfer learned knowledge from one task to another new task, Multi-Task Learning (MTL) in a general sense (Caruana, 1997; Rajpurkar et al., 2016b; Ruder, 2017) has been applied in many fields outside of NLP. Caruana (1993) showed that a model trained in a *multi-task* manner can take advantage of the inductive transfer between tasks, achieving a better generalization performance. MTL has the advantage of computational/storage efficiency (Zhang & Yang, 2017), but training models in a multi-task setting is a balancing act; particularly with datasets that have different: **(a)** dataset sizes, **(b)** task difficulty levels, and **(c)** different types of loss functions. In practice, learning multiple tasks at once is challenging since negative transfer (Wang et al., 2019a), task interference (Wu et al., 2020; Yu et al., 2020) and catastrophic forgetting (Serrà et al., 2018) can lead to worse data efficiency, training stability and generalization compared to single task fine-tuning.

Using Conditionally Adaptive Learning, we seek to improve pretraining knowledge retention and multi-task inductive knowledge transfer. Our contributions are the following:

- A new task conditioned Transformer that adapts and modulates pretrained weights **(Section 2.1)**.
- A novel way to prioritize tasks with an uncertainty based multi-task data sampling method that helps balance the sampling of tasks to avoid catastrophic forgetting **(Section 2.2)**.

Our Conditionally Adaptive Multi-Task Learning (CA-MTL) approach is illustrated in Figure 1. To the best of our knowledge, our work is the first to explore the use of a latent representation of tasks to modularize and adapt pretrained architectures. Further, we believe our work is also the first to examine uncertainty sampling for large-scale multi-task learning in NLP. We show the efficacy of CA-MTL by: **(a)** testing on 26 different tasks and **(b)** presenting state-of-the-art results on a number of test sets as well as superior performance against both single-task and MTL baselines. Moreover, we further demonstrate that our method has advantages over **(c)** other adapter networks, and **(d)** other MTL sampling methods. Finally, we provide ablations and separate analysis of the MT-Uncertainty Sampling technique in section 4.1 and of each component of the adapter in 4.2.

## 2 METHODOLOGY

This section is organized according to the two main MTL problems that we will tackle: (1) How to modularize a pretrained network with latent task representations? (2) How to balance different tasks in MTL? We define each task as: $\mathcal{T}_i \triangleq \{p_i(\mathbf{y}_i|\mathbf{x}_i, \mathbf{z}_i), \mathcal{L}_i, \tilde{p}_i(\mathbf{x}_i)\}$, where $\mathbf{z}_i$ is task $i$'s learnable shallow embedding, $\mathcal{L}_i$ is the task loss, and $\tilde{p}_i(\mathbf{x}_i)$ is the empirical distribution of the training data pair $\{\mathbf{x}_i, \mathbf{y}_i\}$, for $i \in \{1, \ldots, T\}$ and $T$ the number of supervised tasks. The MTL objective is:

$$\min_{\phi(\mathbf{z}),\theta_1,\ldots,\theta_T} \sum_{i=1}^{T} \mathcal{L}_i(f_{\phi(\mathbf{z}_i),\theta_i}(\mathbf{x}_i), \mathbf{y}_i) \tag{1}$$

where $f$ is the predictor function (includes encoder model and decoder heads), $\phi(\mathbf{z})$ are learnable generated weights conditioned on $\mathbf{z}$, and $\theta_i$ are task-specific parameters for the output decoder heads. $\mathbf{z}$ is constructed using an embedding lookup table.

## 2.1 TASK CONDITIONED TRANSFORMER

Our task conditioned Transformer architecture is based on one simple concept. We either add conditional layers or modulate existing pretrained weights using a task representation by extending Feature Wise Linear Modulation (Perez et al., 2018) functions in several ways depending on the Transformer layer. We define our framework below.

**Definition 1** (Conditional Weight Transformations). *Given a neural network weight matrix $\mathbf{W}$, we compute transformations of the form $\phi(\mathbf{W}|\mathbf{z}_i) = \gamma_i(\mathbf{z}_i)\mathbf{W} + \beta_i(\mathbf{z}_i)$, where $\gamma_i$ and $\beta_i$ are learned functions that transform the weights based on a learned vector embedding $\mathbf{z}_i$, for task $i$.*

**Definition 2** (Conditionally Adaptive Learning). *In our setting, Conditionally Adaptive Learning is the process of learning a set of $\phi$s for the conditionally adaptive modules presented below along with a set of task embedding vectors $\mathbf{z}_i$ for $T$ tasks, using a multi-task loss (see equation 1).*

In the subsections that follow: We introduce a new Transformer Attention Module using block-diagonal Conditional Attention that allows the original query-key based attention to account for task-specific biases (section **2.1.1**). We propose a new Conditional Alignment method that aligns the data of diverse tasks and that performs better than its unconditioned and higher capacity predecessor (section **2.1.2**). We adapt layer normalization statistics to specific tasks using a new Conditional Layer Normalization module (section **2.1.3**). We add a Conditional Bottleneck that facilitates weight sharing and task-specific information flow from lower layers (section **2.1.4**). In our experiments we provide an ablation study of these components (Table 1) examining performance in terms of GLUE scores.

### 2.1.1 CONDITIONAL ATTENTION

Given $d$, the input dimensions, the query $\mathbf{Q}$, the key $\mathbf{K}$, and the value $\mathbf{V}$ as defined in Vaswani et al. (2017), we redefine the attention operation:

$$\text{Attention}(\mathbf{Q}, \mathbf{K}, \mathbf{V}, \mathbf{z}_i)) = \text{softmax}\left[M(\mathbf{z}_i) + \frac{\mathbf{Q}\mathbf{K}^T}{\sqrt{d}}\right]\mathbf{V}$$

$$M(\mathbf{z}_i) = \bigoplus_{n=1}^{N} A'_n(\mathbf{z}_i), \quad A'_n(\mathbf{z}_i) = A_n\gamma_i(\mathbf{z}_i) + \beta_i(\mathbf{z}_i)$$

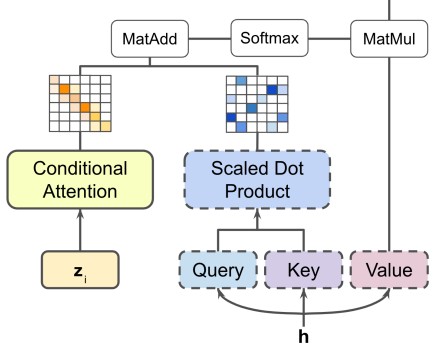

Figure 2: Conditional Attention Module

where $\bigoplus$ is the direct sum operator (see section A.6), $N$ is the number of block matrices $A_n \in \mathbb{R}^{(L/N) \times (L/N)}$ along the diagonal of the attention matrix, $L$ is the input sequence, $M(\mathbf{z}_i) = \text{diag}(A'_1, \dots, A'_N)$ is a block diagonal conditional matrix. Note that $A_n$ is constructed using $L/N$ trainable and randomly initialized $L/N$ dimensional vectors. While the original attention matrix depends on the hidden states $h$, $M(\mathbf{z}_i)$ is a learnable weight matrix that only depends on the task embedding $\mathbf{z}_i \in \mathbb{R}^d$. $\gamma_i, \beta_i : \mathbb{R}^d \mapsto \mathbb{R}^{L^2/N^2}$ are Feature Wise Linear Modulation (Perez et al., 2018) functions. We also experimented with full-block Conditional Attention $\in \mathbb{R}^{L \times L}$. Not only did it have $N^2$ more parameters compared to the block-diagonal variant, but it also performed significantly worse on the GLUE development set (see FBA variant in Table 10). It is possible that GLUE tasks derive a certain benefit from localized attention that is a consequence of $M(\mathbf{z}_i)$. With $M(\mathbf{z}_i)$, each element in a sequence can only attend to other elements in its subsequence of length $L/N$. In our experiments we used $N = d/L$. The full Conditional Attention mechanism used in our experiments is illustrated in Figure 2.

### 2.1.2 CONDITIONAL ALIGNMENT

Wu et al. (2020) showed that in MTL having $T$ separate alignment modules $R_1, \dots, R_T$ increases BERT$_{\text{LARGE}}$ avg. scores on five GLUE tasks (CoLA, MRPC, QNLI, RTE, SST-2) by 2.35%. Inspired by this work, we found that adding a task conditioned alignment layer between the input embedding

layer and the first BERT Transformer layer improved multi-task model performance. However, instead of having $T$ separate alignment matrices $R_i$ for each $T$ task, one alignment matrix $\hat{R}$ is generated as a function of the task embedding $z_i$. As in Wu et al. (2020), we tested this module on the same five GLUE tasks and with BERT$_{\text{LARGE}}$. Enabling task conditioned weight sharing across covariance alignment modules allows us to outperforms BERT$_{\text{LARGE}}$ by 3.61%. This is 1.26 % higher than having $T$ separate alignment matrices. Inserting $\hat{R}$ into BERT, yields the following encoder function $\hat{f}$:

$$\hat{f} = \sum_{t=1}^{T} g_{\theta_i}(E(\mathbf{x}_i)\hat{R}(\mathbf{z}_i)B), \qquad \hat{R}(\mathbf{z}_i) = R\gamma_i(\mathbf{z}_i) + \beta_i(\mathbf{z}_i) \qquad (2)$$

where $\mathbf{x}_i \in \mathbb{R}^d$ is the layer input, $g_{\theta_i}$ is the decoder head function for task $i$ with weights $\theta_i$, $E$ the frozen BERT embedding layer, $B$ the BERT Transformer layers and $R$ the linear weight matrix of a single task conditioned alignment matrix. $\gamma_i, \beta_i : \mathbb{R}^d \mapsto \mathbb{R}^d$ are Feature Wise Linear Modulation functions.

### 2.1.3 CONDITIONAL LAYER NORMALIZATION (CLN)

We extend the Conditional Batch Normalization idea from de Vries et al. (2017) to Layer Normalization (Ba et al., 2016). For task $\mathcal{T}_i$, $i \in \{1, \ldots, T\}$:

$$\mathbf{h}_i = \frac{1}{\sigma} \odot (\mathbf{a}_i - \mu) * \hat{\gamma}_i(\mathbf{z}_i) + \beta_i(\mathbf{z}_i), \qquad \hat{\gamma}_i(\mathbf{z}_i) = \boldsymbol{\gamma}' \gamma_i(\mathbf{z}_i) + \boldsymbol{\beta}' \qquad (3)$$

where $\mathbf{h}_i$ is the CLN output vector, $\mathbf{a}_i$ are the preceding layer activations associated with task $i$, $\mu$ and $\sigma$ are the mean and the variance of the summed inputs within each layer as defined in Ba et al. (2016). Conditional Layer Normalization is initialized with BERT's Layer Normalization affine transformation weights and bias $\boldsymbol{\gamma}'$ and $\boldsymbol{\beta}'$ from the original formulation: $\mathbf{h} = \frac{1}{\sigma} \odot (\mathbf{a} - \mu) * \boldsymbol{\gamma}' + \boldsymbol{\beta}'$. During training, the weight and bias functions of $\gamma_i(*)$ and $\beta_i(*)$ are always trained, while the original Layer Normalization weight may be kept fixed. This module was added to account for task specific rescaling of individual training cases. Layer Normalization normalizes the inputs across features. The conditioning introduced in equation 2.1.3 allows us to modulate the normalization's output based on a task's latent representation.

### 2.1.4 CONDITIONAL BOTTLENECK

We created a task conditioned two layer feed-forward bottleneck layer (CFF up/down in Figure 3). The conditional bottleneck layer follows the same transformation as in equation 2. The module in Figure 3a is added to the top most Transformer layers of CA-MTL$_{\text{BASE}}$ and uses a CLN. For CA-MTL$_{\text{LARGE}}$ this module is the main building block of the skip connection added alongside all Transformer layers seen in Figure 3b. The connection at layer $j$ takes in the matrix sum of the Transformer layer output at $j$ and the previous connection's output at $j - 1$. The Conditional bottleneck allows lower layer information to flow upwards depending on the task. Our intuition for introducing this component is related to recent studies (Tenney

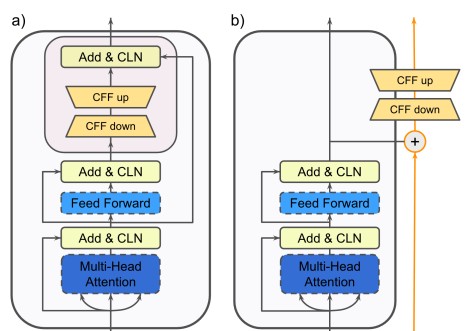

Figure 3: a) Conditional Bottleneck for CA-MTL$_{\text{BASE}}$. b) Conditional Bottleneck for CA-MTL$_{\text{LARGE}}$.

et al., 2019a) that showed that the "most important layers for a given task appear at specific positions". As with the other modules described so far, each task adaptation is created from the weights of a single shared adapter that is modulated by the task embedding.

## 2.2 MULTI-TASK UNCERTAINTY SAMPLING

MT-Uncertainty Sampling is a task selection strategy that is inspired by Active Learning techniques. Our algorithm 1 is outlined in the Appendix, Section A.2. Similar to Active Learning, our algorithm first evaluates the model uncertainty. MT-Uncertainty Sampling uses Shannon Entropy, an uncertainty measure, to choose training examples by first doing forward pass through the model with $b \times T$ input samples. For an output classification prediction with $C_i$ possible classes and probabilities

$(p_{i,1}, \ldots, p_{i,C_i})$, the Shannon Entropy $H_i$, for task $\mathcal{T}_i$ and $i \in \{1, \ldots, T\}$, our uncertainty measure $\mathcal{U}(\mathbf{x})$ are given by:

$$H_i = H_i(f_{\phi(\mathbf{z}_i), \theta_i}(\mathbf{x})) = -\sum_{c=1}^{C_i} p_c \log p_c, \qquad \mathcal{U}(x_i) = \frac{H_i(f_{\phi(\mathbf{z}_i), \theta_i}(\mathbf{x}))}{\hat{H} \times H_i'} \tag{4}$$

$$\hat{H} = \max_{i \in \{1, \ldots, T\}} \bar{H}_i = \max \left[ \frac{1}{b} \sum_{\mathbf{x} \in \mathbf{x}_i} H_i \right], \qquad H_i' = -\sum_{c=1}^{C_i} \frac{1}{C_i} \log \left[ \frac{1}{C_i} \right] \tag{5}$$

where $\bar{H}_i$ is the average Shannon Entropy across $b$ samples of task $t$, $H_i'$, the Shannon entropy of choosing classes with uniform distribution and $\hat{H}$, the maximum of each task's average entropy over $b$ samples. $H_i'$ is normalizing factor that accounts for differing number of prediction classes (without the normalizing factor $H_i'$, tasks with a binary classification $C_i = 1$ were rarely chosen). Further, to limit high entropy outliers and to favor tasks with highest uncertainty, we normalize with $\hat{H}$. The measure in eq. 4 allows Algorithm 1 to choose $b$ samples from $b \times T$ candidates to train the model.

## 3 RELATED WORK

**Multi-Tasking in NLP.** To take advantage of the potential positive transfer of knowledge from one task to another, several works have proposed carefully choosing which tasks to train as an intermediate step in NLP before single task fine-tuning (Bingel & Søgaard, 2017; Kerinec et al., 2018; Wang et al., 2019a; Standley et al., 2019; Pruksachatkun et al., 2020; Phang et al., 2018). The intermediate tasks are not required to perform well and are not typically evaluated jointly. In this work, all tasks are trained *jointly* and *all tasks used* are evaluated from a *single model*. In Natural Language Understanding (NLU), it is still the case that to get the best task performance one often needs a separate model per task (Clark et al., 2019c; McCann et al., 2018). At scale, Multilingual NMT systems (Aharoni et al., 2019) have also found that MTL model performance degrades as the number of tasks increases. We notice a similar trend in NLU with our baseline MTL model. Recently, approaches in MTL have tackled the problem by designing task specific decoders on top of a shared model (Liu et al., 2019b) or distilling multiple single-task models into one (Clark et al., 2019c). Nonetheless, such MTL approaches still involves single task fine-tuning. In this paper, we show that it is possible to achieve high performance in NLU without single task fine-tuning.

**Adapters.** Adapters are trainable modules that are attached in specific locations of a pretrained network. They provide another promising avenue to limit the number of parameters needed when confronted with a large number of tasks. This approach is useful with pretrained MLM models that have rich linguistic information (Tenney et al., 2019b; Clark et al., 2019b; Liu et al., 2019a; Tenney et al., 2019a). Recently, Houlsby et al. (2019) added an adapter to a pretrained BERT model by fine-tuning the layer norms and adding feed forward bottlenecks in every Transformer layer. However, such methods adapt each task individually during the fine-tuning process. Unlike prior work, our method harnesses the vectorized representations of tasks to modularize a single pretrained model across all tasks. Stickland et al. (2019) and Tay et al. (2020) also mix both MTL and adapters with BERT and T5 encoder-decoder (Raffel et al., 2019) respectively by creating local task modules that are controlled by a global task agnostic module. The main drawback is that a new set of non-shared parameters must be added when a new task is introduced. CA-MTL shares all parameters and is able to re-modulate existing weights with a new task embedding vector.

**Active Learning, Task Selection and Sampling.** Our sampling technique is similar to the ones found in several active learning algorithms (Chen et al., 2006) that are based on Shannon entropy estimations. Reichart et al. (2008) and Ikhwantri et al. (2018) examined Multi-Task Active Learning (MTAL), a technique that chooses one informative sample for $T$ different learners (or models) for each $T$ tasks. Instead we choose $T$ tasks samples for *one model*. Moreover, the algorithm weights each sample by the corresponding task score, and the Shannon entropy is normalized to account for various losses (see equation 5). Also, our algorithm is used in a large scale MTL setup ($\gg 2$ tasks). Recently, Glover & Hokamp (2019) explored task selection in MTL using learning policies based on counterfactual estimations (Charles et al., 2013). However, such method considers only fixed stochastic parameterized policies while our method *adapts* its selection criterion based on model uncertainty throughout the training process.

# 4 EXPERIMENTS AND RESULTS

We show that our adapter of section 2 achieve parameter efficient transfer for 26 NLP tasks. Our implementation of CA-MTL is based on HuggingFace (Wolf et al., 2019). Hyperparameters and our experimental set-up are outlined in A.5. To preserve the weights of the pretrained model, CA-MTL's bottom half Transformer layers are frozen in all experiments (except in section 4.4). We also tested different layer freezing configurations and found that freezing half the layers worked best on average (see Section A.8).

## 4.1 MULTI-TASK UNCERTAINTY SAMPLING

Our MT-Uncertainty sampling strategy, from section 2.2, is compared to 3 other task selection schemes: a) Counterfactual b) Task size c) Random. We used a $BERT_{BASE}$ (no adapters) on 200k iterations and with the same hyperparameters as in Glover & Hokamp (2019). For more information on Counterfactual task selection, we invite the reader to consult the full explanation in Glover & Hokamp (2019). For $T$ tasks and the dataset $D_i$ for tasks $i \in \{1, \ldots, T\}$, we rewrite the definitions of Random $\pi_{rand}$ and Task size $\pi_{|task|}$ sampling:

$$\pi_{rand} = 1/T, \quad \pi_{|task|} = |D_i| \left[ \sum_{i=1}^{T} |D_i| \right]^{-1} \quad (6)$$

In Figure 4, we see from the results that MT-Uncertainty converges faster by reaching the 80% average GLUE score line before other task sampling methods. Further, MT-Uncertainty maximum score on 200k iterations is at 82.2, which is 1.7% higher than Counterfactual sampling. The datasets in the GLUE benchmark offers a wide range of dataset sizes. This is useful to test how MT-Uncertainty manages a jointly trained low resource task (CoLA) and high resource task (MNLI). Figure 5 explains how catastrophic forgetting is curtailed by sampling tasks before performance drops. With $\pi_{rand}$, all of CoLA's tasks are sampled by iteration 500, at which point the larger MNLI

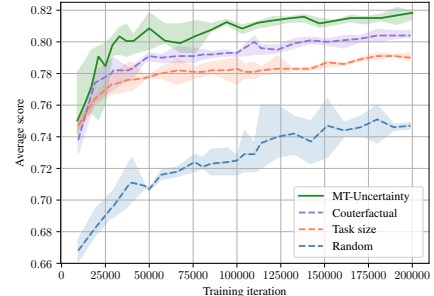

Figure 4: **MT-Uncertainty** vs. other task sampling strategies: median dev set scores on 8 GLUE tasks and using $BERT_{BASE}$. Data for the Counterfactual and Task Size policy $\pi_{|task|}$ (eq. 6) is from Glover & Hokamp (2019).

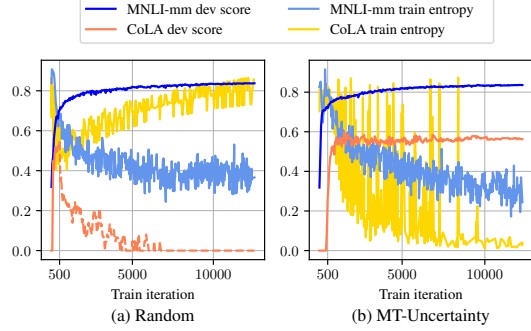

Figure 5: CoLA/MNLI Dev set scores and Entropy for $\pi_{rand}$ (left) and **MT-Uncertainty** (right).

dataset overtakes the learning process and CoLA's dev set performance starts to diminish. On the other hand, with MT-Uncertainty sampling, CoLA is sampled whenever Shannon entropy is higher than MNLI's. The model first assesses uncertain samples using Shannon Entropy then decides what data is necessary to train on. This process allows lower resource tasks to keep performance steady. We provide evidence in Figure 8 of A.2 that MT-Uncertainty is able to manage task difficulty — by choosing the most difficult tasks first.

## 4.2 ABLATION AND MODULE ANALYSIS

In Table 1, we present the results of an ablation study to determine which elements of $CA-MTL_{BERT-BASE}$ had the largest positive gain on average GLUE scores. Starting from a MTL $BERT_{BASE}$ baseline trained using random task sampling ($\pi_{rand}$). Apart for the Conditional Adapter, each module as well as MT-Uncertainty lift overall performance and reduce variance across tasks. Please note that we also included

Table 1: Model ablation study[a] on the GLUE dev set. All models have the bottom half layers frozen.

| Model changes | Avg GLUE | Task $\sigma$ GLUE | % data used |
|---|---|---|---|
| $BERT_{BASE}$ MTL ($\pi_{rand}$) | 80.61 | 14.41 | 100 |
| + Conditional Attention | 82.41 | 10.67 | 100 |
| + Conditional Adapter | 82.90 | 11.27 | 100 |
| + CA and CLN | 83.12 | 10.91 | 100 |
| + MT-Uncertainty ($CA-MTL_{BERT-BASE}$) | **84.03** | **10.02** | 66.3 |

[a]CA=Conditional Alignment, CLN=Conditional Layer Normalization, Task $\sigma$=scores standard deviation *across tasks*.

accuracy/F1 scores for QQP, MRPC and Pearson/ Spearman correlation for STS-B to calculate score standard deviation Task $\sigma$. Intuitively, when negative task transfer occurs between two tasks, either (1) task interference is bidirectional and scores are both impacted, or (2) interference is unidirectional and only one score is impacted. We calculate Task $\sigma$ to characterize changes in the dynamic range of performance across multiple tasks. We do this to asses the degree to which performance improvements are distributed across all tasks or only subsets of tasks. As we can see from Table 1, Conditional Attention, Conditional Alignment, Conditional Layer Normalization, MT-Uncertainty play roles in reducing Task $\sigma$ and increasing performance across tasks. This provides partial evidence of CA-MTL's ability to mitigating negative task transfer.

We show that Conditional Alignment can learn to capture covariate distribution differences with task embeddings co-learned from other adapter components of CA-MTL. In Figure 6, we arrive at similar conclusions as Wu et al. (2020), who proved that negative task transfer is reduced when task covariances are aligned. The authors provided a "covariance similarity score" to gauge covariance alignment. For task $i$ and $j$ with $m_i$ and $m_j$ data samples respectively, and given $d$ dimensional inputs to the first Transformer layer $X_i \in \mathbb{R}^{m_i \times d}$ and $X_j \in \mathbb{R}^{m_j \times d}$, we rewrite the steps to calculate the covariance similarity score between task $i$ and $j$: (a) Take the covariance matrix $X_i^\top X_i$, (b) Find its best rank-$r_i$ approxima-

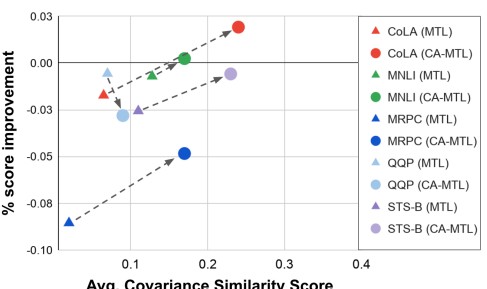

Figure 6: Task performance vs. avg. covariance similarity scores (eq. 7) for MTL and CA-MTL.

tion $U_{i,r_i} D_{i,r_i} U_{i,r_i}^\top$, where $r_i$ is chosen to contain 99% of the singular values. (c) Apply steps (a), (b) to $X_j$, and compute the covariance similarity score $CovSim_{i,j}$:

$$CovSim_{i,j} := \frac{\|(U_{i,r_i} D_{i,r_i}^{1/2})^\top U_{j,r_j} D_{j,r_j}^{1/2}\|_F}{\|U_{i,r_i} D_{i,r_i}^{1/2}\|_F \cdot \|U_{j,r_j} D_{j,r_j}^{1/2}\|_F}. \quad CovSim_i = \frac{1}{T-1} \sum_{j \neq i} CovSim_{i,j} \quad (7)$$

Since we are training models with $T$ tasks, we take the average covariance similarity score $CovSim_i$ between task $i$ and all other tasks. We measure $CovSim_i$ using equation 7 between 9 single-task models trained on individual GLUE tasks. For each task in Figure 6, we measure the similarity score on the MTL trained BERT$_{BASE}$ baseline, e.g., CoLA (MTL), or CA-MTL$_{BERT-BASE}$ model, e.g., MNLI (CA-MTL). Our score improvement measure is the % difference between a single task model and MTL or CA-MTL on the particular task. We find that covariance similarity increases for 9 tasks and that performance increases for 7 out 9 tasks. These measurements confirm that the Conditional Alignment is able to align task covariance, thereby helping alleviate task interference.

### 4.3 JOINTLY TRAINING ON 8 TASKS: GLUE

In Table 2, we evaluate the performance of CA-MTL against single task fine-tuned models, MTL as well as the other BERT-based adapters on GLUE. As in Houlsby et al. (2019), MNLI$_m$ and MNLI$_{mm}$ are treated as separate tasks. Our results indicate that CA-MTL outperforms both the BASE adapter,

Table 2: Adapters with layer freezing vs. ST/MT on GLUE test set. F1 scores are reported for QQP/MRPC, Spearman's correlation for STS-B, accuracy on the matched/mismatch sets for MNLI, Matthew's correlation for CoLA and accuracy for other tasks. * Individual scores not available. ST=Single Task, MTL=Multitask, g.e.= greater or equal to. Results from: [1]Devlin et al. (2018) [2]Stickland et al. (2019). [3]Houlsby et al. (2019) .

| Method | Type | Total params | Trained params/task | # tasks g.e. ST | GLUE | | | | | | | | |
|---|---|---|---|---|---|---|---|---|---|---|---|---|---|
| | | | | | CoLA | MNLI | MRPC | QNLI | QQP | RTE | SST-2 | STS-B | Avg |
| **Base Models — Test Server Results** | | | | | | | | | | | | | |
| BERT$_{BASE}$[1] | ST | 9.0× | 100% | — | 52.1 | 84.6/83.4 | 88.9 | 90.5 | 71.2 | 66.4 | 93.5 | 85.8 | 79.6 |
| BERT$_{BASE}$[2] | MTL | 1.0× | 11.1% | 2 | 51.2 | 84.0/83.4 | 86.7 | 89.3 | 70.8 | 76.6 | 93.4 | 83.6 | 79.9 |
| PALs+Anneal Samp.[2] | MTL | 1.13× | 12.5% | 4 | 51.2 | 84.3/83.5 | 88.7 | 90.0 | 71.5 | 76.0 | 92.6 | 85.8 | 80.4 |
| *CA-MTL*$_{BERT-BASE}$ (ours) | MTL | 1.12× | 5.6 % | 5 | 53.1 | 85.9/85.8 | 88.6 | 90.5 | 69.2 | 76.4 | 93.2 | 85.3 | **80.9** |
| **Large Models — Test Server Results** | | | | | | | | | | | | | |
| BERT$_{LARGE}$[1] | ST | 9.0× | 100% | — | 60.5 | 86.7/85.9 | 89.3 | 92.7 | 72.1 | 70.1 | 94.9 | 86.5 | 82.1 |
| Adapters-256[3] | ST | 1.3× | 3.6% | 3 | 59.5 | 84.9/85.1 | 89.5 | 90.7 | 71.8 | 71.5 | 94.0 | 86.9 | 80.0 |
| *CA-MTL*$_{BERT-LARGE}$ (ours) | MTL | 1.12× | 5.6% | 3 | 59.5 | 85.9/85.4 | 89.3 | 92.6 | 71.4 | 79.0 | 94.7 | 87.7 | **82.8** |

PALS+Anneal Sampling (Stickland et al., 2019), and the LARGE adapter, Adapters-256 (Houlsby et al., 2019). Against single task (ST) models, CA-MTL is 1.3% higher than BERT$_{BASE}$, with 5 out 9 tasks equal or greater performance, and 0.7% higher than BERT$_{LARGE}$, with 3 out 9 tasks equal or greater performance. ST models, however, need 9 models or close to $9\times$ more parameters for all 9 tasks. We noted that CA-MTL$_{BERT-LARGE}$'s average score is driven by strong RTE scores. While RTE benefits from MTL, this behavior may also be a side effect of layer freezing. In Table 10, we see that CA-MTL has gains over ST on more and more tasks as we gradually unfreeze layers.

### 4.4 Transfer to New Tasks

In Table 3 we examine the ability of our method to quickly adapt to new tasks. We performed domain adaptation on SciTail (Khot et al., 2018) and SNLI (Bowman et al., 2015) datasets, using a CA-MTL$_{BASE}$ model trained on GLUE and a new linear decoder head. We

Table 3: Domain adaptation results on dev. sets for *BASE* models. [1]Liu et al. (2019b), [2]Jiang et al. (2020)

| % data used | SciTail | | | | SNLI | | | |
|---|---|---|---|---|---|---|---|---|
| | 0.1% | 1% | 10% | 100% | 0.1% | 1% | 10% | 100% |
| BERT$_{BASE}$[1] | 51.2 | 82.2 | 90.5 | 94.3 | 52.5 | 78.1 | 86.7 | 91.0 |
| MT-DNN[1] | 81.9 | 88.3 | 91.1 | 95.7 | 81.9 | 88.3 | 91.1 | 95.7 |
| MT-DNN$_{SMART}$[2] | 82.3 | 88.6 | 91.3 | **96.1** | 82.7 | 86.0 | **88.7** | **91.6** |
| CA-MTL$_{BERT}$ | **83.2** | **88.7** | **91.4** | 95.6 | **82.8** | **86.2** | 88.0 | 91.5 |

tested several pretrained and randomly initialized task embeddings in a zero-shot setting. The complete set of experiments with all task embeddings can be found in the Appendix, Section A.4. We then selected the best task embedding for our results in Table 3. STS-B and MRPC MTL-trained task embeddings performed best on SciTail and SNLI respectively. CA-MTL$_{BERT-BASE}$ has faster adaptation than MT-DNN$_{SMART}$ (Jiang et al., 2020) as evidenced by higher performances in low-resource regimes (0.1% and 1% of the data). When trained on the complete dataset, CA-MTL$_{BERT-BASE}$ is on par with MT-DNN$_{SMART}$. Unlike MT-DNN$_{SMART}$ however, we do not add context from a semantic similarity model – MT-DNN$_{SMART}$ is built off HNN (He et al., 2019). Nonetheless, with a larger model, CA-MTL surpasses MT-DNN$_{SMART}$ on the full SNLI and SciTail datasets in Table 6.

### 4.5 Jointly training on 24 tasks: GLUE/Super-GLUE, MRQA and WNUT2017

**Effects of Scaling Task Count.** In Figure 7 we continue to test if CA-MTL mitigates *task interference* by measuring GLUE average scores when progressively adding 9 GLUE tasks, 8 Super-GLUE tasks (Wang et al., 2019b), 6 MRQA tasks (Fisch et al., 2019). Tasks are described in Appendix section A.9. The results show that adding 23 tasks drops the performance of our baseline MTL BERT$_{BASE}$ ($\pi_{rand}$). MTL BERT increases by 4.3% when adding MRQA but, with 23 tasks, the model performance drops by 1.8%. The opposite is true when CA-MTL modules are integrated into the model. CA-MTL continues to show gains with a large number of tasks and surpasses the baseline MTL model by close to 4% when trained on 23 tasks.

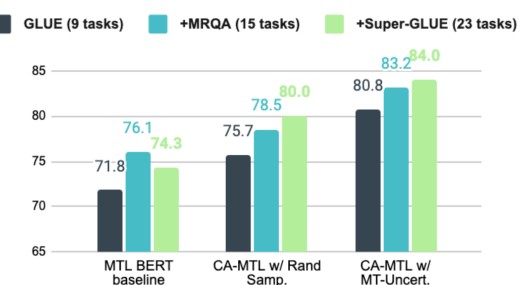

Figure 7: Effects of adding more datasets on avg GLUE scores. Experiments conducted on 3 epochs. When 23 tasks are trained jointly, performance of CA-MTL$_{BERT-BASE}$ continues to improve.

**24-task CA-MTL.** We *jointly* trained large MTL baselines and CA-MTL models on GLUE/Super-GLUE/MRQA and Named Entity Recognition (NER) WNUT2017 (Derczynski et al., 2017). Since some dev. set scores are not provided and since RoBERTa results were reported with a median score over 5 random seeds, we ran our own single seed ST/MTL baselines (marked "ReImp") for a fair comparison. The dev. set numbers reported in Liu et al. (2019c) are displayed with our baselines in Table 9. Results are presented in Table 4.

Table 4: 24-task CA-MTL vs. ST and vs. 24-task MTL with frozen layers on GLUE, SuperGLUE, MRQA and NER development sets. ST=Single Task, MTL=Multitask, g.e.= greater or equal to. Details in section A.5.

| Model | Task Grouping | | | | Avg | # tasks | Total |
|---|---|---|---|---|---|---|---|
| | GLUE | SuperGLUE | MRQA | NER | | e.g. ST | Params |
| *BERT-LARGE models* | | | | | | | |
| ST$_{ReImp}$ | 84.5 | 68.9 | 79.7 | 54.1 | 76.8 | — | 24$\times$ |
| MTL$_{ReImp}$ | 83.2 | 72.1 | 77.8 | 42.2 | 76.4 | 9/24 | 1$\times$ |
| CA-MTL | 86.6 | 74.1 | 79.5 | 49.0 | 79.1 | **17/24** | 1.12$\times$ |
| *RoBERTa-LARGE models* | | | | | | | |
| ST$_{ReImp}$ | 88.2 | 76.5 | 83.6 | 57.8 | 81.9 | — | 24$\times$ |
| MTL$_{ReImp}$ | 86.0 | 78.6 | 80.7 | 49.3 | 80.7 | 7/24 | 1$\times$ |
| CA-MTL | 89.4 | 80.0 | 82.4 | 55.2 | 83.1 | **15/24** | 1.12$\times$ |

We notice in Table 4 that even for large models, CA-MTL provides large gains in performance on average over both ST and MTL models. For the BERT based models, CA-MTL provides 2.3% gain

over ST and higher scores on 17 out 24 tasks. For RoBERTa based models, CA-MTL provides 1.2% gain over ST and higher scores on 15 out 24 tasks. We remind the reader that this is achieved with a single model. Even when trained with 16 other tasks, it is interesting to note that the MTL baseline perform better than the ST baseline on Super GLUE where most tasks have a small number of samples. Also, we used NER to test if we could still outperform the ST baseline on a token-level task, significantly different from other tasks. Unfortunately, while CA-MTL performs significantly better than the MTL baseline model, CA-MTL had not yet overfit on this particular task and could have closed the gap with the ST baselines with more training cycles.

**Comparisons with other methods.** In Table 5, CA-MTL$_{BERT}$ is compared to other Large BERT based methods that either use MTL + ST, such as MT-DNN (Liu et al., 2019b), intermediate tasks + ST, such as STILTS (Phang et al., 2018) or MTL model distillation + ST, such as BAM! (Clark et al., 2019c). Our method scores higher than MT-DNN on 5 of 9 tasks and by 1.0 % on avg. Against STILTS, CA-MTL realizes a 0.7 % avg. score gain, surpassing scores on 6 of 9 tasks. We also show

Table 5: Our 24-task CA-MTL vs. other large models on GLUE. F1 is reported for QQP/MRPC, Spearman's corr. for STS-B, Matthew's corr. for CoLA and accuracy for other tasks. *Split not available. **Uses intermediate task fine-tuning + ST.

| Model | GLUE tasks | | | | | | | | Avg |
|---|---|---|---|---|---|---|---|---|---|
| | CoLA | MNLI | MRPC | QNLI | QQP | RTE | SST-2 | STS-B | |
| *BERT-LARGE based models on Dev set.* | | | | | | | | | |
| MT-DNN | 63.5 | 87.1/86.7 | 91.0 | 92.9 | 89.2 | 83.4 | 94.3 | 90.6 | 85.6 |
| STILTS ** | 62.1 | 86.1* | 92.3 | 90.5 | 88.5 | 83.4 | 93.2 | 90.8 | 85.9 |
| BAM! | 61.8 | 87.0* | – | 92.5 | – | 82.8 | 93.6 | 89.7 | – |
| 24-task CA-MTL | 63.8 | 86.3/86.0 | 92.9 | 93.4 | 88.1 | 84.5 | 94.5 | 90.3 | **86.6** |
| *RoBERTa-LARGE based models on Test set.* | | | | | | | | | |
| RoBERTA** with Ensemble | 67.8 | 91.0/90.8 | 91.6 | 95.4 | 74.0 | 87.9 | 97.5 | 92.5 | **87.3** |
| 24-task CA-MTL | 62.2 | 89.0/88.4 | 92.0 | 94.7 | 72.3 | 86.2 | 96.3 | 89.8 | 85.7 |

that CA-MTL$_{RoBERTa}$ is within only 1.6 % of a RoBERTa ensemble of 5 to 7 models per task and that uses intermediate tasks. Using our 24-task CA-MTL large RoBERTa-based model, we report NER F1 scores on the WNUT2017 test set in Table 6a. We compare our result with RoBERTa$_{LARGE}$ and XLM-R$_{LARGE}$ (Nguyen et al., 2020) the current state-of-the-art (SOTA). Our model outperforms XLM-R$_{LARGE}$ by 1.6%, reaching a new state-of-the-art. Using domain adaptation as described in Section 4.4, we report results on the SciTail test set in Table 6b and SNLI test set in Table 6b. For SciTail, our model matches the current SOTA[2] ALUM (Liu et al., 2020), a RoBERTa large based model that additionally uses the SMART (Jiang et al., 2020) fine-tuning method. For SNLI, our model outperforms SemBert, the current SOTA[3].

Table 6: CA-MTL test performance vs. SOTA.

| **(a) WNUT2017** | F1 |
|---|---|
| RoBERTa$_{LARGE}$ | 56.9 |
| XLM-R$_{LARGE}$ | 57.1 |
| CA-MTL$_{RoBERTa}$ (ours) | **58.0** |

| **(b) SciTail** | % Acc |
|---|---|
| MT-DNN | 94.1 |
| ALUM$_{RoBERTa}$ | 96.3 |
| ALUM$_{RoBERTa-SMART}$ | **96.8** |
| CA-MTL$_{RoBERTa}$ (ours) | **96.8** |

| **(c) SNLI** | % Acc |
|---|---|
| MT-DNN | 91.6 |
| MT-DNN$_{SMART}$ | 91.7 |
| SemBERT | 91.9 |
| CA-MTL$_{RoBERTa}$ (ours) | **92.1** |

## 5 CONCLUSION

We believe that our experiments here have helped demonstrate the potential of task conditioned adaptive learning within a single model that performs multiple tasks. In a large-scale 24-task NLP experiment, CA-MTL outperforms fully tuned single task models by 2.3% for BERT Large and by 1.2% for RoBERTa Large using 1.12 times the number of parameters, while single task fine-tuning approach requires 24 separately tuned single task models or 24 times the number of parameters. When a BERT vanilla MTL model sees its performance drop as the number of tasks increases, CA-MTL scores continue to climb. Performance gains are not driven by a single task as it is often the case in MTL. Each CA-MTL module that adapts a Transformer model is able to reduce performance variances between tasks, increasing average scores and aligning task covariances. This evidence shows that CA-MTL is able to mitigate task interference and promote more efficient parameter sharing. We showed that MT-Uncertainty is able to avoid degrading performances of low resource tasks. Tasks are sampled whenever the model sees entropy increase, helping avoid catastrophic forgetting. Overall, CA-MTL offers a promising avenue to dynamically adapt and modularize knowledge embedded in large monolithic pretrained models. Extending such ideas will be an objective for future work.

---

[2]https://leaderboard.allenai.org/scitail/submissions/public on 09/27/2020
[3]https://nlp.stanford.edu/projects/snli/ on 09/27/2020

ACKNOWLEDGMENTS

This research was supported by the Canada CIFAR AI Chairs Program, NSERC and PROMPT. Experiments in this article were conducted with Compute Canada and MILA computational infrastructure and we thank them for their support. We would like to thank Colin Raffel, Sandeep Subramanian, and Nicolas Gontier for their useful feedback and the anonymous reviewers for helpful comments, discussions and suggestions.

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

# A APPENDIX

## A.1 SUMMARY OF ACRONYMS

Acronyms of datasets and descriptions can be found below in section A.9.

Table 7: List of acronyms used in this paper.

| Acronym | Description |
|---|---|
| ARLM | Autoregressive Language Models |
| CA-MTL | Conditional Adaptive Multi-Task Learning: our architecture |
| CFF | Conditional Feed-Forward: a feed-forward layer modulated by a conditioning vector |
| CLN | Conditional Layer Normalization in section 2.1.3 |
| EDM | Evolutionary Data Measures (Collins et al., 2018): a task difficulty estimate |
| GLUE | General Language Understanding Evaluation Wang et al. (2018): a benchmark with multiple datasets |
| QA | Question Answering |
| MT | Multi-Task |
| MTAL | Multi-Task Active Learning: finding the most informative instance for multiple learners (or models) |
| MLM | Masked Language Model: BERT Devlin et al. (2018) is an example of an MLM |
| MTL | Multi-Task Learning: "learning tasks in parallel while using a shared representation" (Caruana, 1997) |
| MRQA | Machine Reading for Question Answering Fisch et al. (2019): a benchmark with multiple datasets |
| NER | Named Entity Recognition |
| NLP | Natural Language Processing |
| SOTA | State of the art |
| ST | Single Task fine-tuning: all weights are typically updated |
| ST-A | ST with Adapter modules: one adapter per task is trained and pretrained weights are optionally updated |

## A.2 UNCERTAINTY SAMPLING: ALGORITHM AND ADDITIONAL RESULTS

---
**Algorithm 1:** Multi-task Uncertainty Sampling

---
**Input:** Training data $D_t$ for task $t \in [1, \ldots, T]$; batch size $b$; $C_t$ possible output classes for task $t$; $f := f_{\phi(\mathbf{z}_i),\theta_i}$ our model with weights $\phi, \theta_i$;
**Output:** $\mathcal{B}'$ - multi-task batch of size $b$

1  $\mathcal{B} \leftarrow \emptyset$
2  **for** $t \leftarrow 1$ **to** $T$ **do**
3    Generate $\mathbf{x}_t := \{x_{t,1}, \ldots, x_{t,b}\} \overset{\text{i.i.d.}}{\sim} D_t$
4    **for** $i \leftarrow 1$ **to** $b$ **do**
5      $\mathcal{H}_{t,i} \leftarrow -\sum_{c=1}^{C_i} p_c(f(x_{t,i})) \log p_c(f(x_{t,i}))$      ▷ Entropy of each sample
6
7    **end**
8    Compute $\bar{\mathcal{H}}_t \leftarrow \frac{1}{b} \sum_{\mathbf{x} \in \mathbf{x}_i} \mathcal{H}_{t,i}$         ▷ Average entropy for task $t$
9
10   Compute $H'_t \leftarrow -\sum_{c=1}^{C_t} \frac{1}{C_t} \log \left[\frac{1}{C_t}\right]$      ▷ Max entropy (uniform distribution)
11
12   $\mathcal{B} \leftarrow \mathcal{B} \cup \mathbf{x}_t$ and $D_t \leftarrow D_t \setminus \mathbf{x}_t$
13   **if** $D_t = \emptyset$ **then**
14     Reload $D_t$
15   **end**
16   **for** $i \leftarrow 1$ **to** $b$ **do**
17     Compute: $\mathcal{U}_{t,i} \leftarrow \mathcal{H}_{t,i}/H'_t$      ▷ Uncertainty normalized with max entropy
18   **end**
19 **end**
20 Compute $\hat{\mathcal{H}} \leftarrow \max_{i \in \{1,\ldots,T\}}[\bar{\mathcal{H}}_t]$    ▷ Entropy of task with highest average entropy
21 Update $\mathcal{U}_{t,i} \leftarrow \mathcal{U}_{t,i}/\hat{\mathcal{H}}$      ▷ Normalize each sample's uncertainty measure
22 $\mathcal{B}' \leftarrow \text{top\_b}(\{\mathcal{U}_{t,i} | t \in [1, \ldots, T], i \in [1, \ldots, b]\})$    ▷ $b$ samples w/ highest uncertainty
  **Return:** With $\mathcal{B}'$, solve eq. 1 with gradient descent; updated model $f$

---

An advantage of our MT-Uncertainty Sampling approach is its ability to manage task difficulty. This is highlighted in Figure 8. In this experiment, we estimated task difficulty using the Evolutionary Data Measures (EDM)[4] proposed by Collins et al. (2018). The task difficulty estimate relies on multiple dataset statistics such as the data size, class diversity, class balance and class interference. Interestingly, estimated task difficulty correlates with the first instance that the selection of a specific task occurs. Supposing that QNLI is an outlier, we notice that peaks in the data occur whenever tasks are first selected by MT Uncertainty sampling. This process follows the following order: 1. MNLI 2. CoLA 3. RTE 4. QQP 5. MRPC 6.SST-2, which is the order from highest task difficulty to lowest task difficulty using EDM. As opposed to Curriculum Learning (Bengio et al., 2009), MT-Uncertainty dynamically prioritizes the most difficult tasks. As also discovered in MTL vision work (Guo et al., 2018), this type of prioritization on more difficult tasks may explain MT-Uncertainty's improved performance over other task selection methods. In MTL, heuristics to balance tasks during training is typically done by weighting each task's loss differently. We see here how MT-Uncertainty is able to prioritize task difficulty.

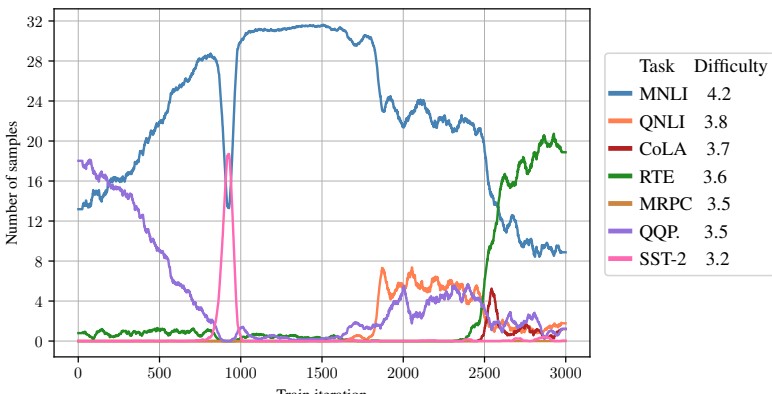

Figure 8: Task composition of MT-Uncertainty sampling and estimated task difficulty using EDM: number of training samples per task at each iteration for batch size of 32. The occurrence of first peaks and estimated difficulty follow the same order: From highest to lowest: MNLI > CoLA > RTE > QQP = MRPC > SST-2.

While the EDM difficulty measure is shown to correlate well with model performance, it lacks precision. As reported in Collins et al. (2018), the average score achieved on the Yahoo Answers dataset is 69.9% and its difficulty is 4.51. The average score achieved on Yelp Full is 56.8%, 13.1% less than Yahoo Answers and its difficulty is 4.42. The authors mention that "This indicates that the difficulty measure in its current incarnation may be more effective at assigning a class of difficulty to datasets, rather than a regression-like value".

## A.3 OTHER RELATED WORK

**Multi-Tasking in NLP and other fields.** MTL weight sharing algorithms such as Mixture-of-Experts (MoE) have found success in NLP (Lepikhin et al., 2020). CA-MTL can complement MoE since the Transformers multi-headed attention can be seen as a form of MoE (Peng et al., 2020). In Vision, MTL can also improve with optimization (Sener & Koltun, 2018) or gradient-based approaches (Chen et al., 2017; Yu et al., 2020).

**Active Learning, Task Selection and Sampling.** Ikhwantri et al. (2018) examined multi-task active learning for neural semantic role labeling in a low resource setting, using entity recognition as the sole auxiliary task. They used uncertainty sampling for active learning and found that 12% less data could be used compared to passive learning. Reichart et al. (2008) has examined different active learning techniques for the two task annotation scenario, focusing on named entity recognition and syntactic parse tree annotations. In contrast, here we examine the larger scale data regime, the modularization of a multi-task neural architecture, and the many task ($\gg 2$) setting among other differences. Other than MTAL (Reichart et al., 2008; Ikhwantri et al., 2018), Kendall et al. (2017) leveraged model uncertainty to balance MTL losses but not to select tasks as is proposed here.

---

[4]https://github.com/Wluper/edm

## A.4 Zero-Shot Results on SciTail and SNLI

Before testing models on domain adaptation in section 4.4, we ran zero-shot evaluations on the development set of SciTail and SNLI. Table 8 outlines 8-task CA-MTL$_{\text{BERT-BASE}}$'s zero-shot transfer abilities when pretrained on GLUE with our MTL approach. We expand the task embedding layer to accommodate an extra task and explore various embedding initialization. We found that reusing STS-B and MRPC task embeddings worked best for SciTail and SNLI respectively.

Table 8: CA-MTL is flexible and extensible to new tasks. However, CA-MTL is sensitive to the new task's embedding. We tested multiple task embeddings that worked best on either SciTail or SNLI by checking performance in a zero shot setting or using 0% of the data.

| Initialization of new task embedding layer | SciTail 0% of data | SNLI 0% of data |
|---|---|---|
| CoLA's embeddings | 43.0 | 34.0 |
| MNLI's embeddings | 24.2 | 33.0 |
| MRPC's embeddings | 34.5 | **45.5** |
| STS-B's embeddings | **46.9** | 33.2 |
| SST-2's embeddings | 25.8 | 34.2 |
| QQP's embeddings | 31.7 | 37.3 |
| QNLI's embeddings | 32.0 | 38.0 |
| RTE's embeddings | 32.3 | 40.6 |
| WNLI's embeddings | 29.0 | 30.4 |
| Average | 28.7 | 37.7 |
| Random initialization | 46.8 | 34.0 |
| Xavier initialization | 29.8 | 37.6 |

## A.5 More Experimental Details

We used a batch size of 32 and a seed of 12 in all experiments. We used Adam (Kingma & Ba, 2015) as the optimizer with a learning rate of 2e-5. We applied a learning rate decay with warm up over the first 10% of the training steps. Unless otherwise specified, we used 5 epochs, a seed of 12 and a sequence length of 128. Additional details are outlined in section . Our data prepossessing and linear decoder heads are the same as in Devlin et al. (2018). We used the same dropout rate of 0.1 in all layers. To run our experiments, we used either four NVIDIA P100 GPU for base models or four NVIDIA V100 GPU for larger ones. We did not perform parameter search. We do not use ensemble of models or task-specific tricks (Devlin et al., 2018; Liu et al., 2019b; Clark et al., 2019c). All models are either 12 Transformer layers for BASE and 24 Transformer layers for LARGE. Apart from CA-MTL, models trained in multi-task learning (BERT or RoBERTa without adapters) used random task sampling. For Table 1 and Figure 7, all BERT-based model have half their layers frozen (untrained) for a fair comparison of ablation results. For the 24-task MTL and CA-MTL models in Tables 4 and 5, we increased the input sequence length to 256 and used 8 epochs.

## A.6 The Direct Sum Operator

In section 2.1.1, we used the direct sum operator $\oplus$. This operation allows us to create a block diagonal matrix. The direct sum of a matrix $A \in \mathbb{R}^{n \times m}$ and $B \in \mathbb{R}^{p \times q}$ results in a matrix of size $(m + p) \times (n + q)$, defined as:

$$\mathbf{A} \oplus \mathbf{B} = \begin{bmatrix} \mathbf{A} & \mathbf{0} \\ \mathbf{0} & \mathbf{B} \end{bmatrix} = \begin{bmatrix} a_{11} & \cdots & a_{1n} & 0 & \cdots & 0 \\ \vdots & \ddots & \vdots & \vdots & \ddots & \vdots \\ a_{m1} & \cdots & a_{mn} & 0 & \cdots & 0 \\ 0 & \cdots & 0 & b_{11} & \cdots & b_{1q} \\ \vdots & \ddots & \vdots & \vdots & \ddots & \vdots \\ 0 & \cdots & 0 & b_{p1} & \cdots & b_{pq} \end{bmatrix}$$

## A.7 BASELINES AND OTHER EXPERIMENTAL RESULTS

In this section, we present our baseline results for BERT, RoBERTa, CA-MTL as well as other models. Our single task results (ST) that we ran ourselves surpass other paper's reported scores in Table 9. Liu et al. (2019c) reports random seed median scores for RoBERTa. However, our RoBERTa ST baseline **matches or surpasses** the original paper's scores **4 out 7 times** on the development set when scores are comparable (QQP F1 and STS-B spearman are not reported).

Table 9: F1 scores are reported for QQP/MRPC, Spearman's correlation for STS-B, accuracy on the matched/mismatch sets for MNLI, Matthew's correlation for CoLA and accuracy for other tasks. ST=Single Task, MTL=Multitask. *QNLI v1 (we report v2) **F1 score or Spearman's correlation is not reported. ***Unknown random seeds. Results from: [1]Stickland et al. (2019) [2]Liu et al. (2019b) [3]Phang et al. (2018) [4]Liu et al. (2019c).

| Method | Total params | Trained params/task | GLUE | | | | | | | | |
|---|---|---|---|---|---|---|---|---|---|---|---|
| | | | CoLA | MNLI | MRPC | QNLI | QQP | RTE | SST-2 | STS-B | Avg |
| **Base Models — Dev set Results** | | | | | | | | | | | |
| PALs+Anneal Samp.[1] | 1.13× | 12.5% | – | – | – | – | – | – | – | – | 81.70 |
| 8-task CA-MTL$_{\text{BERT-BASE}}$ (ours) | 1.12× | 5.6% | 60.9 | 82.7/83.1 | 88.9 | 90.7 | 90.3 | 79.1 | 91.9 | 88.8 | 84.03 |
| **BERT LARGE Models — Dev set Results** | | | | | | | | | | | |
| ST BERT-LARGE[2] | 9× | 100% | 60.5 | 86.7/85.9 | 89.3 | 92.7* | 89.3 | 70.1 | 94.9 | 86.5 | 84.0 |
| ST BERT-LARGE[3] | 9× | 100% | 62.1 | 86.2/86.2 | 92.3 | 89.4 | 88.5 | 70.0 | 92.5 | 90.1 | 84.1 |
| ST BERT-LARGE (ours) | 9× | 100% | 63.6 | 86.5/86.0 | 91.4 | 91.0 | 88.5 | 70.2 | 94.7 | 88.2 | 84.5 |
| 24-task CA-MTL$_{\text{BERT-LARGE}}$ (ours) | 1.12× | 5.6% | 63.8 | 86.3/86.0 | 92.9 | 93.4 | 88.1 | 84.5 | 94.5 | 90.3 | 86.6 |
| **RoBERTa LARGE Models — Dev set Results** | | | | | | | | | | | |
| RoBERTa-LARGE[4] (Median 5 runs)*** | 9× | 100% | 68.0 | 90.2 | 90.9 | 94.7 | ** | 86.6 | 96.4 | ** | – |
| ST RoBERTa-LARGE (ours) | 9× | 100% | 68.3 | 89.2/88.9 | 92.6 | 94.8 | 84.6 | 87.0 | 96.4 | 91.7 | 88.2 |
| 24-task CA-MTL$_{\text{RoBERTa-LARGE}}$ (ours) | 1.12× | 5.6% | 69.7 | 89.4/89.3 | 93.9 | 94.9 | 88.8 | 91.0 | 96.2 | 91.0 | 89.4 |

## A.8 SOME RESULTS ON LAYER FREEZING AND WITH FULL BLOCK ATTENTION.

All experiments in this section were run for only 5 epochs, exclusively on the GLUE dataset for the large BERT-based 8-task CA-MTL model. Results in Table 10 reveal that as we freeze more layers, performance tends to decrease. However, since we wanted to preserve as much pretrained knowledge as possible, we chose to keep at least 50% of layers frozen. While this has slightly lowered our performance on 9 GLUE tasks, we believe that keeping as much of the original pretrained weights is beneficial when increasing the total number of tasks in MTL to 24 or more tasks. However, we did not explore this hypothesis more.

Table 10: **8-task CA-MTL$_{\text{BERT-LARGE}}$** (see section 4.3) for various layer freezing configurations. F1 scores are reported for QQP/MRPC, Spearman's correlation for STS-B, accuracy on the matched/mismatch sets for MNLI, Matthew's correlation for CoLA and accuracy for other tasks. FBA = Full Block Attention

| Method | % frozen layers | # tasks g.e ST | GLUE | | | | | | | | |
|---|---|---|---|---|---|---|---|---|---|---|---|
| | | | CoLA | MNLI | MRPC | QNLI | QQP | RTE | SST-2 | STS-B | Avg |
| **LARGE Models — Dev set Results** | | | | | | | | | | | |
| ST BERT-LARGE (ours) | 0% | — | 63.6 | 86.5/86.0 | 91.4 | 91.0 | 88.5 | 70.2 | 93.1 | 88.2 | 84.3 |
| CA-MTL | 0% | 7 | 60.2 | 86.2/86.0 | 92.0 | 91.5 | 88.7 | 76.3 | 93.3 | 89.5 | 84.9 |
| CA-MTL | 25% | 6 | 63.7 | 86.1/85.8 | 89.1 | 91.2 | 88.6 | 79.7 | 92.9 | 88.5 | 85.1 |
| CA-MTL | 50% | 3 | 63.2 | 85.5/85.5 | 91.8 | 90.9 | 88.3 | 81.4 | 93.0 | 90.1 | 85.5 |
| CA-MTL FBA | 50% | 0 | 60.2 | 81.7/81.1 | 88.0 | 85.8 | 85.7 | 78.7 | 88.6 | 87.1 | 81.8 |

## A.9 DATASET DESCRIPTION

The datasets that were used for the domain adaptation experiments were SciTail[5] and SNLI[6]. We *jointly* trained a CA-MTL$_{\text{RoBERTa-LARGE}}$ model on 9 GLUE tasks, 8 Super-GLUE[7] tasks, 6 MRQA[8] tasks, and on WNUT2017[9] (Derczynski et al., 2017).

All GLUE tasks are binary classification, except STS-B (regression) and MNLI (three classes). We used the same GLUE data preprocessing as in Devlin et al. (2018).

---

[5]https://allenai.org/data/scitail; Leaderboard can be found at: https://leaderboard.allenai.org/scitail/submissions/public

[6]https://nlp.stanford.edu/projects/snli/

[7]https://super.gluebenchmark.com/tasks

[8]https://github.com/mrqa/MRQA-Shared-Task-2019

[9]https://github.com/leondz/emerging_entities_17

Table 11: GLUE (Wang et al., 2018) dataset description.
References: [1]Warstadt et al. (2018), [2]Socher et al. (2013), [3]Dolan & Brockett (2005), [4]Cer et al. (2017), [5]Williams et al. (2018), [6]Wang et al. (2018), [7]Levesque (2011)

| Acronym | Corpus | $|$Train$|$ | Task | Domain |
|---------|--------|-------|------|--------|
| CoLA[1] | Corpus of Linguistic Acceptability | 8.5K | acceptability | miscellaneous |
| SST-2[2] | Stanford Sentiment Treebank | 67K | sentiment detection | movie reviews |
| MRPC[3] | Microsoft Research Paraphrase Corpus | 3.7K | paraphrase detection | news |
| STS-B[4] | Semantic Textual Similarity Benchmark | 7K | textual similarity | miscellaneous |
| QQP | Quora Question Pairs | 364K | paraphrase detection | online QA |
| MNLI[5] | Multi-Genre NLI | 393K | inference | miscellaneous |
| RTE[6] | Recognition Textual Entailment | 2.5K | inference/entailment | news, Wikipedia |
| WNLI[7] | Winograd NLI | 634 | coreference | fiction books |

Table 12: Super-GLUE (Wang et al., 2019b) dataset description. References: [1]Clark et al. (2019a), [2]de Marneffe et al. (2019), [3]Gordon et al. (2012), [4]Khashabi et al. (2018), [5]Zhang et al. (2018), [6]Wang et al. (2019b), [7]Poliak et al. (2018), [8]Levesque (2011)

| Acronym | Corpus | $|$Train$|$ | Task | Domain |
|---------|--------|-------|------|--------|
| BoolQ[1] | Boolean Questions | 9.4K | acceptability | Google queries, Wikipedia |
| CB[2] | CommitmentBank | 250 | sentiment detection | miscellaneous |
| COPA[3] | Choice of Plausible Alternatives | 400 | paraphrase detection | blogs, encyclopedia |
| MultiRC[4] | Multi-Sentence Reading Comprehension | 5.1K | textual similarity | miscellaneous |
| ReCoRD[5] | Reading Comprehension and Commonsense Reasoning | 101K | paraphrase detection | news |
| RTE[6] | Recognition Textual Entailment | 2.5K | inference | news, Wikipedia |
| WiC[7] | Word-in-Context | 6K | word sense disambiguation | WordNet, VerbNet |
| WSC[8] | Winograd Schema Challenge | 554 | coreference resolution | fiction books |

Table 13: MRQA (Fisch et al., 2019) dataset description. References: [1]Rajpurkar et al. (2016a), [2]Trischler et al. (2017), [3]Joshi et al. (2017), [4]Dunn et al. (2017), [5]Yang et al. (2018), [6]Kwiatkowski et al. (2019)

| Acronym | Corpus | $|$Train$|$ | Task | Domain |
|---------|--------|-------|------|--------|
| SQuAD[1] | Stanford QA Dataset | 86.6K | crowdsourced questions | Wikipedia |
| NewsQA[2] | NewsQA | 74.2K | crowdsourced questions | news |
| TriviaQA[3] | TriviaQA | 61.7K | trivia QA | web snippets |
| SearchQA[4] | SearchQA | 117.4K | Jeopardy QA | web snippets |
| HotpotQA[5] | HotpotQA | 72.9K | crowdsourced questions | Wikipedia |
| Natural Questions[6] | Natural Questions | 104.7K | search logs | Wikipedia |

SuperGLUE has a more diverse task format than GLUE, which is mostly limited to sentence and sentence-pair classification. We follow the same preprocessing procedure as in Wang et al. (2019b). All tasks are binary classification tasks, except CB (three classes). Also, WiC and WSC are span based classification tasks. We used the same modified MRQA dataset and preprocessing steps that were used in Joshi et al. (2019). All MRQA tasks are span prediction tasks which seeks to identify start and end tokens of an answer span in the input text.

Table 14: SNLI (Bowman et al., 2015) and SciTail (Khot et al., 2018) datasets description.

| Acronym | Corpus | $|$Train$|$ | Task | Domain |
|---------|--------|-------|------|--------|
| SNLI[1] | Stanford Natural Language Inference | 550.2k | inference | human-written English sentence pairs |
| SciTail[2] | Science and Entailment | 23.5K | entailment | Science question answering |

SNLI is a natural inference task where we predict three classes. Examples of three target labels are: Entailment, Contradiction, and Neutral (irrelevant). SciTail is a textual entailment dataset. The hypotheses in SciTail are created from multiple-choice science exams and the answer candidates (premise) are extracted from the web using information retrieval tools. SciTail is a binary true/false classification tasks that seeks to predict whether the premise entails the hypothesis. The two datasets are used only for domain adaptation in this study (see section A.4 for the details of our approach).

