# OpenReview forum: "Conditionally Adaptive Multi-Task Learning: Improving Transfer Learning in NLP Using Fewer Parameters & Less Data"
_ICLR.cc/2021/Conference — ICLR 2021 Poster_

### Official Review · AnonReviewer1 · 2020-10-26
**A good paper on multitask learning that proposes novel task embedding-grounded architectural modules and task-sensitive sampling with good results and thorough experiments, analyses and ablations**

**Rating:** 8
**Confidence:** 4

**Review:**

**Summary and Key Contributions:**

This paper proposes the use of task embeddings for a novel conditional attention mechanism and various task-conditioned modules. The paper also proposes the use of an entropy-based multi-task uncertainty sampling to automatically determine how to sample the data for training the MT architecture. The paper demonstrates the effectiveness of the proposed architecture and sampling through multiple comprehensive experiments and ablations/analyses.

**Positives:**

* The paper proposes several modifications to the transformer architecture that enable leveraging a learned task-specific embedding, namely conditional attention, conditional alignment, conditional layer norm and conditional adapter.
* The paper proposes the use of the entropy-based MT-Uncertainty Sampling to choose which training examples to use, in a manner that accounts for some of the subtleties (that were perhaps not immediately obvious, but clearly visible in hindsight), such as accounting for the differing numbers of classes in each task.
* Most of the modifications proposed are clean, well motivated and make a lot of intuitive sense. The overall approach also seems to work well without the need for extensive hyperparameter tuning.
* Strong, thorough experimental validation of the proposed techniques. Evaluations on GLUE, Super-GLUE show the proposed method often outperforming (or at least performing comparably to) several strong baselines, both MT-based and single model/fine-tuned, and with fewer parameters. The paper's proposed approach sets a new state-of-the-art performance on 3 tasks (WNUT-2017, SciTail, SNLI).
* Good ablations of the utility of the different task embedding-grounded modules, on the performance of MT-Uncertainty sampling compared to other methods, how MT-Uncertainty sampling tends to choose most difficult tasks first while also avoiding catastrophic forgetting, and of transferring to new tasks or progressively adding in additional tasks.

**Suggestions:**
* The paper claims to attain 2.2% higher performance compared to a full fine-tuned BERT large model (abstract line 16). However, it seems like the paper compares a CA-MTL-RoBERTa-Base model to a BERT-base baseline. This is a somewhat unfair comparison, and it might be better to either clarify or to compare the baseline to the CA-MTL-BERT-Base model (which based on Table 2 would still be a rather impressive 1.3% better while making the comparison fairer).
* While I understand that space is limited, Section 2.1 might have benefited from explaining FiLM, its use here and some intuition, to help keep the paper more self-contained.

**Questions/Clarifications:**
* (A5 Table 11) On SciTail, it appears that Random Init performs almost as well as STS-B in the zero-shot setting, which is rather surprising. Do you have any intuition/analysis as to why this might be the case?
* (line 104-105) How exactly was the block matrix A_n obtained? And does $\bigoplus$ represent the diag operator?
* (line 95) Does $p_i(\mathbf{y_i}|\mathbf{x_i},\mathbf{z_i})$ represent the model prediction score, i.e., $f_{\phi(\mathbf{z_i}), \theta_i}(\mathbf{x_i})$?
* In Table 5, the total data used was 66.3% (or 64.6%). How was the amount of data to be used determined? Was a stopping point determined based on the GLUE dev set?
* The paper refers to "assemble methods" (line 267, Table 4 left header, line 657). Was it perhaps meant to say "ensemble"?

=================================

**Update:**

Thank you for you responses and clarifications to mine and the other reviewers' comments. Thank you also for baking so much of the reviewers' feedback into your latest draft, in particular by improving how the comparisons are presented in the abstract and the rest of the paper, by updating Table 2 to make it clearer and cleaner, and by adding experiments and analyses in Section 4. As a suggestion pertaining to the latest draft, I agree with R5 that a consequence of the additions to Section 4 have made it a little crammed, although I certainly understand that the constrained spaced forced the authors to make this trade-off. Overall, I continue to maintain that this is a good paper with novel ideas that are well-justified by experimental results and analyses, and would like to reiterate that I believe that this work is a clear accept.

---

> ### Author Response · Authors · 2020-11-21
> **Thank you for your interesting questions. We also used your feedback to improve the paper.**
>
> Thank you for taking the time to read and review our paper. We are very pleased that you have found our work interesting.
>
> It is true that results were not always easy to align with the appropriate baselines. We have simplified the tables by removing RoBERTa from table 2. This will facilitate comparisons with BERT based models. We ran our own baselines and have added comparisons with our approach with single-task fine-tuned and MTL RoBERTa models in Table 4. In Table 5, we compare our method to published numbers. Against the BERT Large intermediate task fine-tuning method STITLS, CA-MTL realizes a 0.7 % avg. score gain, surpassing scores on 6 out of 9 tasks.  Our method scores higher than BERT Large MT-DNN, another method that uses MTL, on 5 out of 9 tasks and by 1.0 % on average.
>
> Thank you for suggesting that “Section 2.1 might have benefited from explaining FiLM, its use here and some intuition, to help keep the paper more self-contained”. We will add details to the appendix soon.
>
>
> Regarding your questions:
> - (A5 Table 12): We agree that it is worth investigating why the 8-task CA-MTL model (based on GLUE) with a random init performed so well in the zero-shot setting on SciTail. SciTail is a dataset composed of science questions. The domain is substantially different from the domain of other GLUE tasks. In fact, when we try the same zero-shot experiments with our 24-task CA-MTL model, we find that random init drops to a value that is closer to the average init. MRQA may contain data that is closer to SciTail. However, the best zero-shot embedding is chosen outside of MRQA task embeddings. We would need to dig further to understand that particular behavior.
> - The ⨁ symbol does indeed represent the diagonal operator.
> - In pytorch, A_n is created with the following code snippet:
> M = torch.block_diag(*list(out_b.chunk(N, 0))) // N is the number of blocks
> We apply FiLM on a randomly initialized and learnable weight matrix using task_embedding:
> A_n = nn.parameters(L, L/N)
> A’_n = FiLM(A_n, task_embedding)
> // Then we chunk out_b in sub matrices of size L/N x L/N with chunk(N, 0)
> // We use torch.block_diag to place the sub matrices along the diagonal
>
> - “Does pi(yi|xi,zi) represent the model prediction score, i.e., fϕ(zi),θi(xi)?” it is and we will make sure to make that clearer. Thank you for your question.
> - The amount of data to be used is not a hyperparameter that we choose. The model chooses a data sample on which to train based on the uncertainty measure of equation 4. Our sampling algorithm is able to track samples that are used for backpropagation during training. Each sample from a different task has a unique ID. At the end of training, we are able to identify which samples were not used for backpropagation throughout the training process. We can then calculate the % of data used at any specific training iteration.
> - Thank you for pointing out the typo. We have done more proof-reading.

---

### Official Review · AnonReviewer2 · 2020-10-28
**Interesting and Important Work, but Confusing Presentation and Evaluation**

**Rating:** 6
**Confidence:** 3

**Review:**

Update: After seeing the changes the authors have made, I have slightly increased my score. Thanks!

The authors investigate how to use multi-task training on top of a pretrained model for a variety of unrelated tasks from the GLUE benchmark. Their goal is to reduce the amount of parameters and data needed, while improving (or maintaining) state-of-the-art performance.

The main contribution are the following innovations: a special attention mechanism called block-diagonal conditional attention, a set of modules for adaptation of a pretrained model, and an uncertainty-based multi-task data sampling method. The latter aims at avoiding catastrophic forgetting, while the former avoid having to share all (or no) parameters.

Experiment are performed on the GLUE benchmark. The authors already published their code in an anonymous GitHub repository (that's great!).

Overall, I like the idea of the paper: it is important to reduce the parameters needed for sets of tasks, to enable NLP models to be deployed in a larger variety of settings, and reducing the amount of data needed. Also, the authors did a great job in terms of conducting evaluations from different angles. For instance, they ask how different amounts of data influence model behavior or if their data sampling method can react to task difficulty.

Unfortunately, I think this particular paper might need some more work before being ready for publication. I will discuss my concerns one-by-one in detail:
- Most importantly, the evaluation is confusing. The authors switch between using BERT_base and RoBERTa_base without being very clear about when and why, e.g., in Table 2 they have both models, but only in different sections. This makes a direct comparison impossible. Also in Table 2, they claim that they outperform other approaches. However, they compare to BERT models and build themselves on RoBERTa_base; how are the results meaningful if they use a stronger model to start with? (And the comparison to their BERT_base-based model yields roughly equal scores.) I compared their numbers explicitly to Liu et al. (2019), and RoBERTa_base outperforms their approach on nearly all tasks (and on average). I would argue that this is not a fair comparison: did the authors intentionally choose weak baselines? The same applies to Table 3: it is unclear to me why or how the baseline T5 model has been chosen. I do understand that the main point is the reduction of the amount of parameters (per task), but this doesn't mean that the evaluation should paint a wrong picture.
- Additionally, in my opinion, the authors are misrepresenting prior work when saying in line 163 that the "MTL approach has not yet been successful in NLP". The authors refer to the fact that MTL can (and often does!) lead to performance decrease for individual tasks. However, this is not the case, for instance, for massively multilingual MT (see the reference below for an example). Granted, the final effect of MTL depends on task similarities, but that's probably the same for the proposed approach.
- Major contributions of the work should be described in the main paper. Therefore, it's not ideal that the uncertainty sampling algorithm has been moved to the appendix.

Additionally, there are some minor things I would add or improve:
- I would add references to multi-task training on different languages (e.g., Task 1 is translation from EN to FR and Task 2 is translation from EN to DE). For instance: Roee Aharoni, Melvin Johnson, and Orhan Firat. 2019. Massively multilingual neural machine translation. NAACL. (And the references therein.)
- The paper would benefit from more proofreading. For instance, line 94: "task latent" -> "latent task".

---

> ### Author Response · Authors · 2020-11-21
> **We have updated our tables with the appropriate baselines to make comparisons and evaluation more intuitive.**
>
> Thank you for taking the time to read and review our paper. It was encouraging to see that you found our paper to be interesting and important work.
>
> We have changed the presentation of the results to make it easier to compare similar methods with each other. We have removed the RoBERTa result from Table 2 and we included RoBERTa baselines in Table 4 and Table 5. Also, we made sure that our BERT models are compared to other BERT based MTL approaches. Since some development set scores are not provided and since RoBERTa results were reported with a median score over 5 random seeds, we have run our own single seed ST/MTL baselines (marked “ReImp”) for a fair comparison in Table 4. The development set numbers reported in Liu et al. (2019c) are displayed with our RoBERTa baselines in Table 13. In Table 5, we compare our method to the published numbers. Against the BERT Large intermediate task fine-tuning method STITLS, CA-MTL realizes a 0.7 % avg. score gain, surpassing scores on 6 out of 9 tasks.  Our method scores higher than BERT Large MT-DNN, another method that uses MTL, on 5 out of 9 tasks and by 1.0 % on average.
>
> - Regarding your comment "MTL approach has not yet been successful in NLP", we have modified this statement and written the following: “In Natural Language Understanding (NLU), it is still the case that to get the best task performance one often needs a separate model per task (Clark et al., 2019c; McCann et al., 2018).”
> - Due to the space limitation and the added analysis, we are forced to keep our sampling algorithm in the appendix. However, we feel that the additional explanations and discussions bring forward the key contributions of our MT-Uncertainty sampling algorithm.
> - As per your recommendation, we have added references to Aharoni, Melvin Johnson, and Orhan Firat. 2019. Massively multilingual neural machine translation. NAACL.
> -We have also done more proof reading. Thank you for catching the typo.

---

### Official Review · AnonReviewer4 · 2020-11-01
**Well written paper on NLP Multitask learning, could improve motivation and connection between MTL challenges and modeling choices**

**Rating:** 6
**Confidence:** 3

**Review:**

This paper proposed a new transformer model framework for multitask learning on NLP tasks. To deal with challenges in multitask learning/co-training, the authors proposed five improvements, including modifications on the transformer layers with task conditioning and uncertainty sampling. In the experiments, the authors showed that the proposed model can outperform full fine-tuned BERT large model with less parameters (adding some parameters on a single co-trained model), and with less training data.

This paper is very well written and easy to follow. It is also a very meaningful application as pretrain-finetuning based approach could be both sub-optimal and ineffiicent.

The proposed model structure and sampling techniques suits well for multitask learning based solutions. Having soft-parameter sharing conditioned by task embeddings while freezing part of pre-trianed parameters seems reasonable and efficient. The uncertainty based sampling approach is also very interesting and could potentially benefit other multitask learning problems in general.

The authors also conducted experiments on various types of NLP tasks including GLUE and Super-GLUE, compared with baseline methods such as pretrain finetuning BERT model, adapter based approach, and T5.

However, I think the paper could improve in the following aspects.
A1. In-depth connection to different multitask learning challenges. This paper proposed 5 modeling components. Intuitively, they all target at solving multitask learning challenges mentioned earlier in this paper, e.g., capacity balancing on multiple tasks, catastrophic forgetting, negative task transfer and interference. I think more in-depth discussion on how each of the 5 modeling improvements solve what aspects of those challenges can singficanlty improve the impact of this paper.

For example, the conditional alignment part, inspired by Wu et al. (2020), could have some similar discussion related to the covariate shifts among different tasks. Uniquely, it is less clear such alignment matrix can learn to capture covariate distribution difference with task embeddings co-learned from other model improvement components introduced in this paper.

Similarly, it would be interesting to see discussions on how the uncertainty sampling can improve what aspects in multitask learning. Comparing to many sampling tricks (e.g., discussed in T5), would uncertainty sampling dynamically improve task capacity balancing? Either empirical analysis or theoretical justification could improve this subsection.

It would also be helpful to understand how these 5 model improvements interact with each other, both intuitively and empirically.

A2. Might need to compare and/or include some existing work. I see two lines of related work are missing.

(1) There are similar research in NLP mutltiask learning that adopts mixture-of-expert based model architecture to do soft-parameter sharing, for example:
[1] https://arxiv.org/pdf/2006.16668.pdf
[2] https://arxiv.org/pdf/2007.05891.pdf
And routing/mixute-of-expert based task conditioning is somewhat popular in other domains. It might be interesting to compare, or to point out the modeling/motivation differences of the proposed work compared to those approaches.

(2) Dynamic reweighting based optimization algorithms, for example:
[1] https://arxiv.org/abs/1711.02257
[2] https://arxiv.org/abs/1705.07115
[3] https://arxiv.org/abs/1810.04650
I think this line of work might be less popular in NLP based tasks but overall would be meaningful to be included in a discussion.

a minor comments:
It might be good to explicitly mention each of the proposed improvements in figure 1, for example, conditional attention, conditional layer norm, ..

---

> ### Author Response · Authors · 2020-11-21
> **Section 4: Improved motivation and connection between MTL challenges and modeling choices.**
>
> We appreciate your comments and thoughtful recommendations to improve the connection between MTL and our proposed method.
>
> We agree that in-depth discussion on the effects of each module on multitask learning challenges is warranted. With the extra page, we have restructured the manuscript to begin with the analysis of CA-MTL Adapter modules and the MT-Uncertainty sampling strategy. This is a summary by section aligned with specific MTL challenges (copied from the new version):
> - Section 4.1: We study our MT Uncertainty Sampling vs other task sampling methods (such as Counter Factual task selection) on a baseline model without adapters. We also show how MT Uncertainty helps avoid catastrophic forgetting.
> - Section  4.2: we analyze covariate shift and study ablations of CA-MTL modules. We observe higher average scores and lower score variance, revealing that CA-MTL helps mitigate negative task transfers. Input embeddings after Conditional Alignment exhibit improved task covariance similarity.
> - Section 4.4: we provide a simple method to ``reconfigure CA-MTL's weights on a new task using task embeddings which facilitates more efficient knowledge transfer.
> - Section 4.5: we investigate large scale MTL on 24 tasks. CA-MTL exhibits higher performance with increased task count, demonstrating its ability to better balance model capacity at scale.
>
>
> Thank you for suggesting Wu et al. (2020)’s covariance shift analysis. We have added Figure 6 that shows that our Conditional Alignment layer increases the covariance alignment score between one task and the other remaining tasks, which helps increase performance. Following the theoretical justification of Wu et al. (2020), we can conclude that our alignment module also helps decrease negative task transfer and interference.
>
> Thank you for suggesting the papers in A2. We have added them to the related works and have described the differences between CA-MTL and other methods. We have also added a few clarifications in figure 1. We will soon add a larger figure with more details in the appendix.

---

### Official Review · AnonReviewer5 · 2020-11-06
**A multi-task learning method that outperforms other Adapters-style approaches, but some claims are overly generous.**

**Rating:** 7
**Confidence:** 4

**Review:**

Update: see comment below for rationale behind change from 5 to 7.

## Claimed Contribution:

This paper seeks to develop a multi-task learning (MTL) strategy that performs competitively with single-task (ST) fine-tuning despite having fewer parameters per task. To this end, this work introduces several task-specific modules components and an uncertainty-weighted sampling strategy.
	Some of the task-specific module components are similar to the Adapters line of work, with a key difference being that the adapters are trained in a MTL setting versus in ST finetuning. The authors also introduce latent representations of tasks that feed into these different modules. The sampling strategy is well motivated and credibly improves results.
	Overall, this MTL approach can(*) match the average performance of single-task finetuning without having to keep a separate copy of the parameters for each task.

(*) see caveats in Cons.

### Pros

* Paper is well-written, concepts are introduced clearly, related work section is well-done.
* Comparisons are made to relevant related work, including relatively recent papers.
* Ablations in Table 5 are very useful and convincing.
* Method perform well compared to alternatives in the Adapters world.
* Figure 5 is also helpful to understand the benefits of incremental tasks.
* The sampling method is interesting and compelling, and appendix A2 gives additional interesting results. This strikes me as an important contribution that deserves to be explored more widely, including outside of the Adapters context.


### Cons

* Paper tends to overclaim / have some comparisons that don’t have the necessary context. When correcting for those claims, the results are awash with its ST-tuned alternatives and worse than the MTL-then-ST counterparts. In particular:
  * The abstract claim “With our base model, we attain 2.2% higher performance compared to a full fine-tuned BERT large model on the GLUE benchmark, adding only 5.6% more trained parameters per task” has a  typo that makes it less impressive, as shown in Table 2. This gap is for a full fine-tuned BERT base model, not large. The large BERT model outperforms the CA-MTL base ones by 0.3. In addition, the authors compare a BERT checkpoint to a Roberta one, which feels like something that should be mentioned upfront.
  * I am not a fan of only exposing the #params per task in Table 3 and 4. I would like to also see the row on #parameters of the model on each task (~220M for T5, ~370 for CA-MTL). It is worth mentioning in Table 3 that you are comparing a BERT-large encoder only model to a ~T5-base encoder decoder. The choice of SpanBERT in Table 4 is also suboptimal, as you are comparing a BERT checkpoint (albeit with an additional QA focused pre-training step) to a Roberta one.
  * Table 4 compares models derived from BERT checkpoints to yours that is derived from a Roberta one, this should be emphasized more clearly as it explains some of the score discrepancy.
* There are no comparisons to single-task finetuned Roberta models. On the Dev set, these outperform by 0.7 on Glue for the Base models (see https://arxiv.org/pdf/1907.11692.pdf Table 8), though the story varies considerably by task.
* There is little mention made of the discrepancies across tasks. For the same base model, CA-MTL underperforms on many tasks, recovering most of the points on RTE, a task that is known for performing well when doing MTL with MNLI (see for instance STILTs, Phang et al, 2018, who find a double digit point gain on Dev when using a BERT model finetuned on MNLi before finetuning on RTE). I think wider discussion of the discrepancy across task is warranted. Overall, it seems like CA-MTL can capture part of the gains of MTL+ST tuning but still underperform ST tuning on tasks where little is gained by doing MTL.
* The STILTs paper ( https://arxiv.org/abs/1811.01088 ) seems relevant at least in the related work, esp. considering you are citing some of its follow-up papers. It might also be worth comparing to models that do intermediate fine-tuning.
* There is little discussion of when this would be advantageous. Since the results are often lower than the ST and MTL-then-ST counterparts, it would be worth emphasizing that the lower disk space requirements are key in mobile/on-device contexts.
* One redeeming remark for some of those points: the papers this method compares to often explore a wider hyperparam grid for each task, while this method does not.

### Recommendation

Overall, I think this is an interesting approach. However, some of the claimed results are not as strong when put in the proper context, which the authors sometime fail to do. As such, I think this paper is a slight reject in its current state. With a fairer presentation of the comparisons it might be a slight accept.

---

> ### Author Response · Authors · 2020-11-21
> **More baselines that align between model types and size as well as better comparisons to other MTL.**
>
> We appreciate you taking the time to read our paper and to write a well-structured and detailed review.
>
> We streamlined the presentation of the results to make it easier to compare similar methods with each other.  We removed the RoBERTa results from Table 2 and we included RoBERTa baselines in Table 4 and Table 5. Also, we made sure that our BERT models are compared to other BERT based MTL approaches. Assessing our performance against SpanBERT was indeed suboptimal. We made sure to have better comparisons.
>
> We will address specific comments:
>
> - In our previous version, we stated “with our base model, we attain 2.2% higher performance”. This statement was not clear that we were measuring the relative % change. Also, we were not specific enough about which quantities we were comparing. We now compare models of the same size and type.
>
> - “It is worth mentioning in Table 3 that you are comparing a BERT-large encoder only model to a ~T5-base encoder decoder”: In the older version, the reference that points to T5 says “baseline in Raffel et al. (2019a)”. We made it clearer and specified that T5 is an encoder-decoder model in Section 3.
>
> - Following your request, we have added the total parameters for all models in Table 14. Also, we have added “total params” columns when compared with Adapters in Table 2 and when compared with other MLT-BERT-based approaches in Table 5. In Table 4, we now have a RoBERTa single-task fine-tuned models for all tasks. We also specified that the ALUM (the previous SOTA on SciTail) in table 6b is a Large RoBERTa-based model. We also brought the NER table, which already has comparisons with RoBERTa-based models, from the appendix to the main body (table 6a).
>
> - “CA-MTL underperforms on many tasks, recovering most of the points on RTE”. It is true that RTE performs better in an MTL setting. However, in table 2, CA-MTL BERT-BASE performs as well or better than a BERT-BASE single-task fine-tuned on 5 out 9 scores (see the “e.g. ST” column) and is 1.3% higher on average. Compared to MTL BERT-BASE from Stickland et al. (2019), our method performs better on 6 out 9 scores and 1.0% on average. For the 24-task Large model, we notice the same trend in Tables 4 and 5. The Large 8-task model’s dominance on RTE scores in Table 2 is explained away as we unfreeze layers (see Table 15).
>
> - As per your suggestion, we have added the comparison to other MTL or intermediate task fine-tuning methods such as STILTs, BAM! and MT-DNN to Table 5.
>
> Thank you for reminding us the importance of the lower disk space requirements. We have added this detail in section 3. Also, we appreciate your comment: “one redeeming remark for some of those points: the papers this method compares to often explore a wider hyperparam grid for each task, while this method does not”.

---

> > ### Comment · AnonReviewer5 · 2020-11-23
> > **Score increased from 5 to 7: Welcome revisions with more adequate comparisons. Paper a bit crammed, hopefully can be improved in later versions/camera-ready.**
> >
> > The authors have made welcome changes to the paper, making many comparisons more adequate. In particular, I feel like the distinction in the paper between Roberta and Bert checkpoints is clearer. Reading the other reviews and replies, I believe the comments made by other reviewers have also been addressed and I particularly welcome the new presentation of Section 4.
> >
> > Due to these improvements and space constraints, the paper has become a bit crammed in pages 7-9. I would encourage the authors to try to resolve this in an eventual camera-ready version.
> >
> > These improvements have prompted me to increase the score from 5 to 7. The presentation of the experiments is now more fair and the paper as a whole has substantially improved (though differences in hyperparam-tuning across some methods still make some comparisons hard).
> >
> > Other new remarks:
> > * Appendix 7 is incomplete: "Results in Figure 15 reveal that..."
> > * When comparing the MT Uncertainty sampling, it might be worth to also try $Task size ^ \alpha$ where $\alpha = 0.5$ or $\alpha = 0.75$. This is often used as a compromise sampling strategy for MTL. Not urgent but would be welcome in a camera-ready version.
> >
> > Some other things I caught:
> > * Table 4 legend: g.e is explained but not used in table.
> > * "Comparison with other methods" STILTS is typo'ed as STITLS once.
> > * In Appendix A5 you mention twice that you use the seed 12 (great choice!)

---

### Official Review · AnonReviewer3 · 2020-11-10
**A collection of strategies to improve multitask learning and bring performance on par with single task training.**

**Rating:** 6
**Confidence:** 4

**Review:**

The paper explores a collection of strategies to improve multitask learning and bring performance on par with single task training. The strategies build on and show good awareness of existing work and achieve positive overall results on GLUE, SuperGLUE, and MRQA tasks. Abolation experiments demonstrate the value of the individual components being proposed. The results are particularly impressive given the small number of additional parameters introduced into the model and in the context of prior work that has shown mixed results from multitask training.

However, despite the strong results, I still have mixed feelings about this paper. The presentation follows the format of introducing a fairly large number of new methods followed by experiments that directly attempt to exceed the current SoTA. The analysis of the contribution of the individual methods, while greatly appreciated, still almost seems like an after thought. To be a strong accept for ICLR or another similar venue, it think the techniques explored in this paper deserve an alternative treatment that focus on the contribution of the individual methods first and then concludes with their combination. With this presentation, it would be good to include results for any non-standard design choices. For example, on line 110, you have "We also experimented with full-block 110
Conditional Attention ∈ R L×L, Not only did it have N2 more parameters compared to the block-diagonal variant". I'd like to see the numbers that support this claim.

I also found some of the comparisons in the paper somewhat confusing. For example, Table 2 compares a BERT baseline to the authors proposed bag of methods, dubbed CA-MTL, combined with a RoBERTa model. If these numbers are to be meaningful, it would be good to include a RoBERTa baseline in the tables.

Finally, one of the original motivations for multitask training was that it might allow us to make use of less data for each individual task. While it's great that the paper shows good performance on GLUE using a fraction of the available training data, and good domain adaption with varying amounts of training data, it would be desirable to dig a little deeper. It would be interesting to see something like Table 1 for other tasks being explore in the paper of perhaps using the whole GLUE benchmark (e.g., using less than 64% of the training data and finding the knee where performance drops off).

---

> ### Author Response · Authors · 2020-11-21
> **Improved results presentation + additional analysis of the adapter and the MT-Uncertainty sampling.**
>
> Thank you for taking the time to read our paper and for your useful comments.
>
> Following your recommendation, we streamlined the presentation of the results to make have cleaner comparisons of methods. We removed the RoBERTa result from Table 2 and we included RoBERTa baselines in Table 4 and Table 5.
>
> We agree that a detailed study of the contribution of the individual modules is required. With the extra page, we added more analysis of each method’s contribution (Section 4.1 and 4.2). We brought forward discussions of the ablation study (Table 1). We also added experimental results of the full-block Conditional Attention performance on GLUE (line 110 of the older version) in Table 15.
>
> Following your specific comment:
>
> - “It would be interesting to see something like Table 1 for other tasks being explored in the paper'' - First, note that  Table 1 is now Table 3. We explored varying data sizes at the task level as shown in Figure 7. We used GLUE only (39% of the tasks), GLUE and MRQA (65% of the tasks) and GLUE, MRQA and SuperGlue (100% of the tasks) for both MTL and CA-MTL models. To make this point more obvious, we have added details of this experiment in the discussion (section 4.5). Please note that the percentage of the data used was not a limit that we set; rather, it was selected by our sampling algorithm during training.

---

### Decision · Program_Chairs · 2021-01-07
**Final Decision**

**Decision:**

Accept (Poster)

**Comment:**

I thank the authors for their constructive engagement in the review process - this paper clearly benefitted from your prompt attention to initial weaknesses - and I thank the reviewers for their excellent, detailed, careful reviews.

This paper presents methods that allow a multi-task learning (MTL) system to performs competitively against single-task (ST) fine-tuning despite using fewer parameters and less data per task. This is a great goal, which disrupts the currently most pursued approach of ST fine-tuning and beats aiming for fully single model like BAM!

Pros

* The paper presents a number of useful methods, some novel like conditional attention mechanisms, for improving MTL. There is a lot in this paper.
* Clean, motivated, intuitive model modifications
* Comprehensive experiments
* Generally well-written paper, fairly comprehensive discussion of related work

Cons

The initial paper had a number of weaknesses, the most consistently mentioned being that the results presented tend to overclaim and to confuse (flipping between BERT and RoBERTa, etc.). The authors have done a good job revising the paper to address these concerns but should certainly remember these points in producing the final version.

The paper is a bit of a grab bag of small contributions that together help for MTL, but which isn't as strong a story as one big novel idea carefully presented and evaluated. While this paper has a ton of work behind it, and a lot of good stuff in it, I think this does somewhat weigh against the paper being selected for spotlight presentation.

Requests for the final version:

Further small clean up: e.g., still one "STITLS" to become "STILTS", some cleaning up could be done in the references where "stilts" is lowercase, there isn't capitalization after punctuation in the title of either Clark et al. paper, and the de Marneffe et al. paper, Fisch et al. paper and others lack information on where the work appeared).

Despite the improvements, I think more can and should be done to make the paper clearer and better laid out. Here are a few suggestions:

* More consistency in labeling the contributions might be possible. I'm imagining that they might be able to be consistently labeled 1 to 5, rather than being 1 to 5 in subsections of section 2, but 1 to 3 with bulleted sub items in the introduction, and several of them (a) through (c) in Fig 1.
* Although the reviewers generally said the paper was clear and well-written, I still feel that section 2.1 is harder to get than it should be – and indeed you resorted to pasting PyTorch code in this discussion to help us! While people often say that a figure should be standalone, maybe that doesn't really make sense when it's an inline figure like this, and you'd be better off explaining things well once (mainly in the text) rather than badly twice?!? I.e., just make the Fig 2 caption: "Figure 2: Conditional attention". I still think it's harder to get what the equations are doing than it should be, because there is inadequate text explaining the equations and the ideas motivating them. Although you now sort of sneak in to the text that ⨁ means diag, you still don't explicitly say so. I think having even 2 more lines of text at your disposal could really help if well deployed.
* As R5 observes, the compactification in pages 7-9 just gets kind of ridiculous. I get that you're trying to deal with page breaks and to fit a lot in etc., but it just makes no sense when the tables embedded into paragraphs on p9 don't even belong with the corresponding paragraph! How might this be fixed? It's difficult, since you do just have a lot of material. The best idea I can come up with would be to shorten the main paper related work section to only 1 paragraph that discusses things at a very high level, and to put your detailed related work (even adding a few more of the things reviewers suggest like MT, etc.) to the Appendix. This might give you another half page in the main paper to make things more sensible.